# Vascular remodeling is governed by a VEGFR3-dependent fluid shear stress set point

Nicolas Baeyens[1], Stefania Nicoli[1], Brian G Coon[1], Tyler D Ross[1], Koen Van den Dries[1], Jinah Han[1], Holly M Lauridsen[2], Cecile O Mejean[1], Anne Eichmann[1], Jean-Leon Thomas[3,4,5,6], Jay D Humphrey[2], Martin A Schwartz[1,2,7]*

[1]Department of Internal Medicine, Yale Cardiovascular Research Center, Yale University School of Medicine, New Haven, United States; [2]Department of Biomedical Engineering, Yale University School of Engineering and Applied Science, New Haven, United States; [3]Department of Neurology, Yale University School of Medicine, New Haven, United States; [4]Université Pierre et Marie Curie, Paris, France; [5]INSERM, CNRS U-1127, UMR-7225, Paris, France; [6]PHP, Groupe Hospitalier Pitié Salpêtrière, Paris, France; [7]Department of Cell Biology, Yale University School of Medicine, New Haven, United States

*For correspondence: martin. schwartz@yale.edu

Competing interests: The authors declare that no competing interests exist.

**Abstract** Vascular remodeling under conditions of growth or exercise, or during recovery from arterial restriction or blockage is essential for health, but mechanisms are poorly understood. It has been proposed that endothelial cells have a preferred level of fluid shear stress, or 'set point', that determines remodeling. We show that human umbilical vein endothelial cells respond optimally within a range of fluid shear stress that approximate physiological shear. Lymphatic endothelial cells, which experience much lower flow in vivo, show similar effects but at lower value of shear stress. VEGFR3 levels, a component of a junctional mechanosensory complex, mediate these differences. Experiments in mice and zebrafish demonstrate that changing levels of VEGFR3/Flt4 modulates aortic lumen diameter consistent with flow-dependent remodeling. These data provide direct evidence for a fluid shear stress set point, identify a mechanism for varying the set point, and demonstrate its relevance to vessel remodeling in vivo.

## Introduction

Homeostasis, one of the central concepts in physiology (*Cannon, 1929*), posits that physiological variables have an optimum value or set point such that deviations from that set point activate responses that return those variables toward their original value. For example, changes in central body temperature trigger sweating, altered blood flow to the skin or shivering to restore normal temperature. In the vasculature, arteries remodel under sustained changes in blood flow, with increased or decreased flow triggering outward or inward remodeling, respectively, to adjust lumen diameters accordingly (*Thoma, 1893*; *Kamiya and Togawa, 1980*; *Kamiya et al., 1984*; *Langille and O'Donnell, 1986*; *Langille et al., 1989*; *Langille, 1996*; *Tronc et al., 1996*; *Tuttle et al., 2001*). These studies have given rise to the concept that the endothelium encodes a fluid shear stress set point that governs remodeling responses (*Rodbard, 1975*; *Cardamone and Humphrey, 2012*) (*Figure 1A*). While appealing, there is no direct evidence for such a mechanism. Moreover, if it exists, the set point must itself be variable, since different types of vessels, for example, arteries, veins and lymphatics,

**eLife digest** Blood and lymphatic vessels remodel their shape, diameter and connections during development, and throughout life in response to growth, exercise and disease. This process is called vascular remodeling.

The endothelial cells that line the inside of blood and lymphatic vessels are constantly exposed to the frictional force from flowing blood, termed fluid shear stress. Changes in shear stress are sensed by the endothelial cells, which trigger vascular remodeling to return the stress to the original level. It has been proposed that remodeling is governed by a preferred level of fluid shear stress, or set point, against which deviations in the shear stress are compared. Thus, changing the fluid flow through a blood vessel increases or decreases shear stress, which results in the vessel remodeling to restore the original level of shear stress. Like all remodeling, this process involves inflammation to recruit white blood cells, which assist with the process.

Baeyens et al. investigated whether such a shear stress set point exists and what its biological basis might be using cultured endothelial cells from human umbilical veins. These cells remained stable and in a resting state when a particular level of shear stress was applied to them; above or below this shear stress level, the cells produced an inflammatory response like that seen during vascular remodeling. This suggests that these cells do indeed have a set point for shear stress. The same response occurred in human lymphatic endothelial cells, although in these cells the shear stress set point was much lower, correlating with the low flow in lymphatic vessels.

Baeyens et al. then discovered that the shear stress set point is related to the level of a protein called VEGFR3 in the cells, which was recently found to participate in shear stress sensing. Endothelial cells from lymphatic vessels normally produce much greater quantities of VEGFR3 than those from blood vessels. Reducing the amount of VEGFR3 in lymphatic endothelial cells increased the set point shear stress, while increasing the levels in blood vessel cells decreased the set point. This suggests that the levels of this protein account for the difference in the response of these two cell types. Baeyens et al. then tested this pathway by reducing the levels of VEGFR3 in zebrafish embryos and in adult mice. In both animals, this caused arteries to narrow, showing that VEGFR3 levels also control sensitivity to shear stress—and hence vascular remodeling—inside living creatures.

Understanding in detail how vascular remodeling is regulated could help improve treatments for a wide range of cardiovascular conditions. To do so, further work will be needed to develop methods to control the sensitivity of endothelial cells to shear stress and to identify other proteins that might specifically control the narrowing or the expansion of vessels in human patients.

generally have very different magnitudes of fluid shear stress (*Lipowsky et al., 1980*; *Dixon et al., 2006*; *Suo et al., 2007*).

Arterial remodeling is crucial in normal physiological adaptation to growth and exercise, and is a major determinant of outcomes in cardiovascular disease (*Kohler et al., 1991*; *Corti et al., 2011*; *Padilla et al., 2011*). Outward remodeling of atherosclerotic vessels helps to maintain lumen diameter and blood flow, whereas inward remodeling leads to ischemia associated with angina and peripheral vascular disease (*Ward et al., 2000*). Additionally, flow-dependent remodeling of small blood vessels near sites of myocardial infarction provides collateral circulation that plays a major role in restoring cardiac function (*Heil and Schaper, 2004*), whereas failure to remodel is a major factor in progression to heart failure.

Flow-dependent remodeling is initiated by inflammatory activation of the endothelium, leading to recruitment of leukocytes that assist with remodeling in several ways including secretion of matrix metalloproteinases, cytokines and extracellular matrix proteins (*Silvestre et al., 2008*; *Schaper, 2009*; *Silvestre et al., 2013*). Once the remodeling phase is completed, inflammation is resolved and the vascular wall stabilized. NF-κB plays a major role in the initial inflammatory activation (*Castier et al., 2009*; *Sweet et al., 2013*), whereas signaling through TGF-β is critical in the anti-inflammatory, stabilization phase (*Walshe et al., 2009*) .

These considerations led us to investigate the existence of a fluid shear stress set point and its relevance to vascular remodeling. Our results provide strong evidence for a fluid shear stress set point

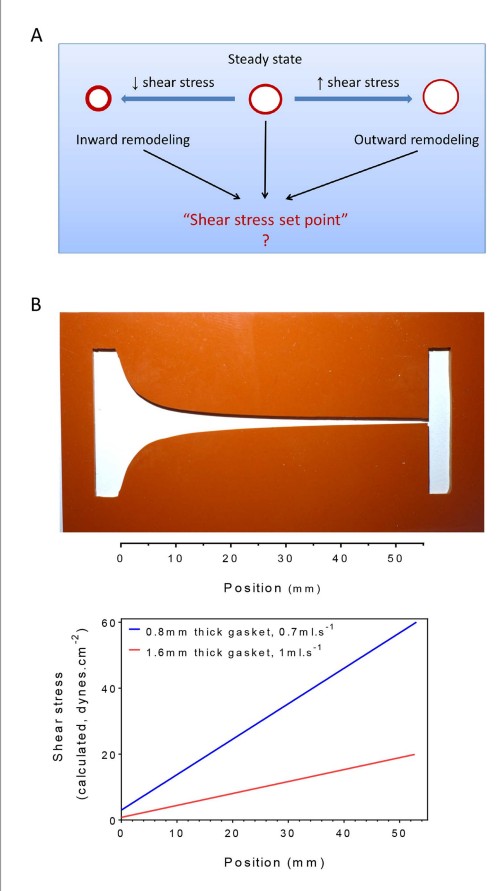

**Figure 1**. Testing the set point hypothesis. (**A**) Definition of the 'shear stress set point'. (**B**) Picture of a silicone gasket used in the gradient flow chamber with the corresponding calculation of the theoretical shear stress level across the channel with two different conditions of gasket thickness and flow rate.

in vascular endothelium. They also show that vascular and lymphatic endothelium have different set points, that this difference is mediated by differences in expression of VEGFR3, and provide evidence that this pathway controls artery remodeling in vivo.

## Results

### Is there a set point for fluid shear stress?

To test the existence of a shear stress set point, we built a flow chamber that creates a gradient of shear stress along a single culture slide. Following a previous design (*Usami et al., 1993*), the width of the chamber progressively decreases to yield a linear gradient (*Figure 1B*). We then measured several biological responses associated with fluid shear stress and vascular remodeling. To assay responses as a function of shear stress, we took successive microscopic images along the chamber. Depending on localization, these responses correlated with calculated values of shear stress. Changing the gasket thickness and flow rate allowed us to control the range of shear stress for each experiment (*Figure 1B*).

We first measured endothelial cell alignment in flow, which is a well-studied response associated with vessel stabilization and suppression of inflammatory pathways (*Levesque and Nerem, 1985*; *Wang et al., 2012*; *Baeyens et al., 2014*). Alignment was quantified by measuring the angle between the major axis of the nucleus and the flow direction (*Baeyens et al., 2014*). Human umbilical vein endothelial cells (HUVECs) were subjected to 16 hr of laminar shear stress ranging from 2 to 60 dynes.cm$^{-2}$. HUVECs aligned in the direction of the flow, between

approximately 10 and 20 dynes.cm$^{-2}$, but were misaligned or oriented perpendicularly, against the flow direction, outside this range (*Figure 2A*, *Figure 2—figure supplement 1*). This result agrees with previous studies showing perpendicular alignment of endothelial cells under very high shear stress (*Viggers et al., 1986*; *Dolan et al., 2011*; *Dolan et al., 2012*).

Next, to assess NF-κB activation, we measured the nuclear translocation of the p65 subunit of NF-κB. NF-κB showed baseline activation in cells without flow, which decreased between approximately 10 and 25 dynes.cm$^{-2}$, and dramatically increased at very high shear (*Figure 2B*, *Figure 2—figure supplement 1*). The suppression of NF-κB translocation in this range is consistent with previous observations that sustained laminar flow is anti-inflammatory (*Mohan et al., 1997*; *Berk, 2008*). Lastly, we measured the activation of TGFβ/SMAD signaling by assaying nuclear translocation of Smad1. Strikingly, flow induced Smad translocation with a sharp maximum between 10 and 20 dynes.cm$^{-2}$ and repressed translocation at higher values (*Figure 2C*, *Figure 2—figure supplement 1*). The results obtained with the gradient chamber were validated by examining 2, 12 and 50 dynes.cm$^{-2}$ using normal parallel flow chambers (*Figure 2—figure supplement 1*).

Taken together, these results show that HUVECs have a biphasic response to shear stress such that anti-inflammatory, stabilization pathways are activated between approximately 10 and 20 dynes.cm$^{-2}$, while lower or higher shear stress is pro-inflammatory. This behavior is consistent with a shear stress set point within the range of 10 and 20 dynes.cm$^{-2}$ for these cells.

## Analysis of lymphatic endothelial cells

An essential aspect of the set point hypothesis is that it must differ between different types of vessels. In vivo, average shear stress in lymphatic vessels is much lower than in arteries or veins (*Lipowsky et al., 1980*; *Dixon et al., 2006*; *Suo et al., 2007*). We therefore examined the behavior of human dermal lymphatic endothelial cells (HDLEC), using modified chamber parameters to obtain values of shear stress from 0.5 to 20 dynes.cm$^{-2}$ (*Figure 1*). In these experiments, HUVECs aligned between 8 and 20 dynes.cm$^{-2}$, (*Figure 2A* and *Figure 3A*) whereas HDLEC aligned maximally between 4 and 6 dynes.cm$^{-2}$ (*Figure 3A*, *Figure 3—figure supplement 1*). The minimum for NF-κB translocation also shifted to between 4 and 10 dynes.cm$^{-2}$ (*Figure 3B*, *Figure 3—figure supplement 1*), which corresponds well to in vivo measurements (*Dixon et al., 2006*). These results indicate that lymphatics have a higher sensitivity to shear stress compared to HUVECs, consistent with the set point concept.

## VEGFR3 expression regulates the set point for shear stress in vitro

A number of shear stress responses, including cell alignment and NF-κB activation, require mechanotransduction via VEGFR2, whose ligand-independent transactivation by flow requires PECAM-1 and VE-cadherin (*Tzima et al., 2005*). We therefore considered whether differences in expression of these proteins might account for the difference in flow sensitivity between HUVECs and HDLECs. However, no major differences in levels of these proteins were observed (*Figure 3C*). VEGFR3, a close homolog of VEGFR2, is highly expressed in lymphatic cells (*Kaipainen et al. 1995*) and recent work in our lab showed that it is activated by flow in vascular endothelial cells similarly to VEGFR2 (*Coon et al., 2015*). These considerations prompted us to examine levels of this receptor as well, which showed approximately 10-fold higher expression in lymphatic ECs compared to HUVECs (*Figure 3C*). We therefore considered whether VEGFR3 levels might be responsible for the higher flow sensitivity of lymphatic ECs.

HDLECs were therefore transfected with VEGFR3 siRNA, which reduced its expression to approximate the level in HUVECs (*Figure 4A*). We also transduced HUVECs with adenovirus coding for hVEGFR3-GFP (*Figure 4A*), which increased levels by ~10-fold and infected >90% of the cells (*Figure 4—figure supplement 1*). Cell alignment in flow was then analyzed. Depletion of VEGFR3 in HDLECs shifted the optimal alignment to between 10 to 20 dynes.cm$^{-2}$ (*Figure 4B*, *Figure 4—figure supplement 2*), similar to HUVECs. Conversely, over-expression of VEGFR3 in HUVECs decreased the optimal response toward the lower shear stress levels seen with lymphatic ECs (*Figure 4C*, *Figure 4—figure supplement 2*). Taken together, these results show that VEGFR3 levels are a major determinant of the difference in shear stress sensitivity between HUVECs and HDLECs.

We also confirmed VEGFR3 activation by flow in lymphatic endothelial cells. Onset of flow stimulated VEGFR3 phosphorylation maximally at 6 dynes.cm$^{-2}$ in HDLEC (*Figure 5*), which corresponds well to the set point of around 5 dynes.cm$^{-2}$ in these cells. HUVECs, by contrast, exhibited a weaker response that was shifted to higher shear, consistent with their higher set.

## VEGFR3 controls blood vessels diameter in zebrafish, in a VEGF-C-independent manner

To test whether VEGFR3 levels control sensitivity to shear stress and vascular remodeling in vivo, we examined *Danio rerio* (zebrafish). This system has the advantage that development proceeds normally without blood flow, thus, fluid shear stress can be altered or even stopped without affecting viability (*Langheinrich et al., 2003*). The notion that levels of VEGFR3 (Flt4 in zebrafish) determine the shear stress set point predicts that reducing VEGFR3 expression will induce inward remodeling of the vessels in order to increase shear stress and restore normal signaling. We used a strain in which blood vessels are labeled by expression of *kdrl:mCherry* (VEGFR2) and *flt4:Citrine* (VEGFR3) reporters. *kdrl:mCherry* was highly visible in the dorsal aorta and the posterior cardinal vein, whereas *flt4:Citrine* was low (though detectable) in the dorsal aorta and higher in the cardinal posterior vein and the developing thoracic duct (*Figure 6*, *Figure 6—figure supplement 1*). Flt4/VEGFR3 and its ligand, VEGF-C, are associated with development of lymphatic vasculature and segmental arteries in zebrafish (*Covassin et al., 2006*; *Kuchler et al., 2006*). To assay the effect of FLT4 and VEGFC dosage on vessels diameter, we injected zebrafish embryos at the one cell stage with previously validated VEGFC and FLT4 morpholinos at two different concentrations. These antisense oligos target the respective mRNAs and induce a dose dependent loss of function (*Nicoli et al., 2012*;

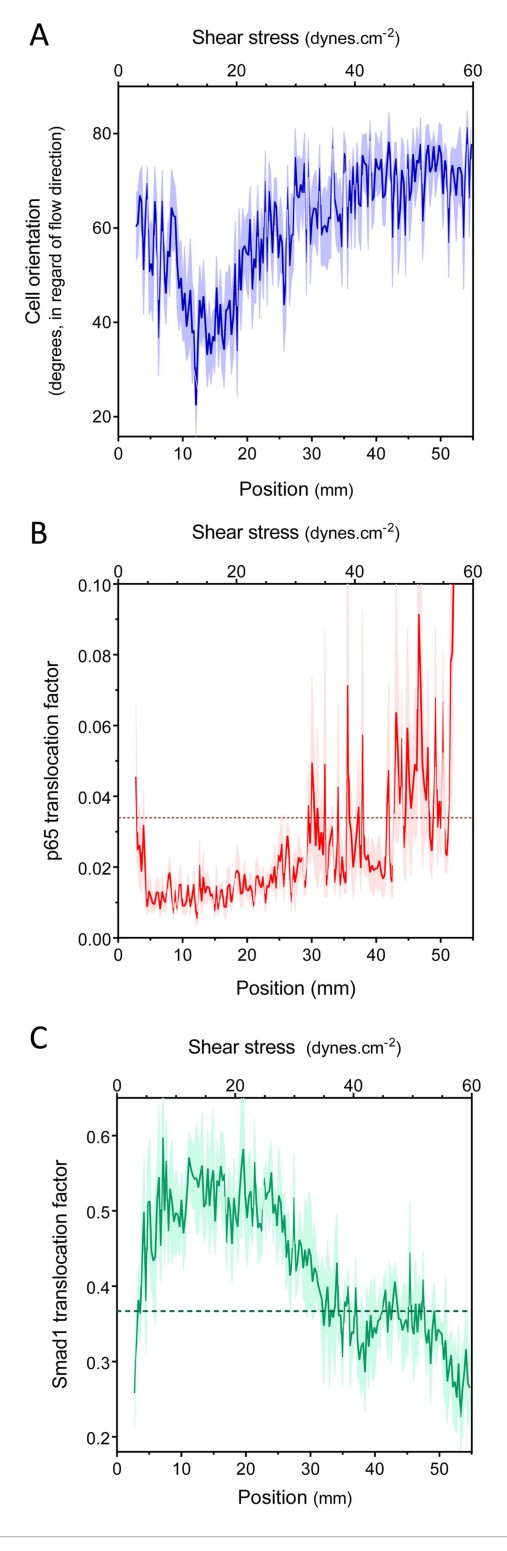

**Figure 2**. Set point for shear stress. (**A**) Cell orientation: the average orientation of HUVEC nuclei was measured in each picture, to obtain average orientation at a given shear stress. (n = 16, Mean ± SEM, ANOVA: F = 15.02, p < 0.0001). With no flow, cell orientation was random (average = 45˚). (**B**) NF-κB activation: p65 nuclear
*Figure 2. continued on next page*

*Villefranc et al., 2013*). At 72 hr post fertilization (hpf), the progressive inhibition of VEGFC did not perturb the remodeling of blood vessel or vessel diameter but as expected inhibited the development of the thoracic duct, the first zebrafish lymphatic vessel (*Yaniv et al., 2006*) (*Figure 6*, white stars). By contrast, progressive inhibition of FLT4 reduced the diameter of the dorsal aorta with loss of thoracic duct evident at a higher dose of FLT4 morpholino (*Figure 6*). These results suggested that VEGF-C-independent Flt4 activation is required for artery diameter and exclude an indirect effect of lymphatic development on the artery development. Interestingly, a similar decrease of the dorsal aorta diameter can be observed in a recent paper (*Kwon et al., 2013*). Although these authors focused on the growth of motoneurons, the dorsal aorta is readily visible in images of Flt1 mCherry reporter embryos; its diameter is obviously smaller in *expando* embryos expressing a kinase dead Flt4, as well as in wildtype embryos treated with Flt4 morpholino or VEGFR3 inhibitors but not after injection with VEGFC morpholino, in accordance with our own observations.

To test the role of flow in this process, embryos were treated with 40 μM nifedipine, a voltage-dependent calcium channel blocker that stops the heart and thus blood flow (*Langheinrich et al., 2003*). Blocking flow led to a decreased vessel diameter (*Figure 6*, *Figure 6—figure supplement 1*), supporting the role of shear stress in determining lumen size. Interestingly, lumen diameter was similar in embryos treated with high dose Flt4 morpholino and with nifedipine. To test whether Flt4 acts on a flow pathway, we then combined these treatments. Strikingly, in the absence of flow, neither Flt4 nor VEGF-C morpholinos caused further changes in vessel size. Taken together, these results support the conclusion that VEGF-C-independent activation of VEGFR3 by flow may determine the endothelial cell sensitivity to flow and vessel remodeling, consistent with the existence of a fluid shear stress set point.

Interestingly, ligand-independent responses for VEGFR3 are consistent with developmental mouse phenotypes: deletion of VEGF-C and VEGF-D does not affect the development and maturation of blood vessels during mice development, while deletion of VEGFR3 does (*Haiko et al., 2008*). Ligand-dependent responses are thus required for lymphangiogenesis but probably not for flow responses.

*Figure 2. Continued*

translocation in HUVEC was measured either in no flow (dotted line: average) or after 16 hr of flow in the gradient chamber (n = 6, Mean ± SEM, ANOVA: F = 10.97, p < 0.0001). (**C**) Smad1 activation: Smad1 nuclear translocation in HUVECs was measured without flow (dotted line: average) or after 16 hr of flow in the gradient chamber (n = 6, Mean ± SEM, ANOVA: F = 13.47, p < 0.0001).

The following figure supplement is available for figure 2:

**Figure supplement 1**. (**A**) Quantification of cell orientation, p65 nuclear translocation or smad1 nuclear translocation without flow or after 16 hr laminar flow at the indicated values (NS: not significant, *: p < 0.05, **: p < 0.01, ****: p < 0.0001).

## VEGFR3 and artery remodeling in mice

Lastly, we investigated whether VEGFR3 controls artery remodeling in mice in a similar manner. Expression of VEGFR3 in adult arteries has been reported to be low (*Gu et al., 2001*; *Witmer et al., 2002*; *Tammela et al., 2008*), thus, we first verified its transcription in the thoracic aorta. Using a transgenic *Vegfr3::YFP* reporter mouse (*Calvo et al., 2011*), expression of YFP was readily detected, confirming *Vegfr3* expression in adult arteries (*Figure 7A*). We confirmed this observation by staining a longitudinal section of the thoracic aorta with an anti-VEGFR-3 antibody (*Figure 7B*). Interestingly, VEGFR3 expression was not uniform: weaker expression was detected in the outer curvature or some portions of the carotid artery, associated with higher shear stress, while stronger expression was observed in the inner curvature, associated with low shear stress (*Figure 7—figure supplement 1*).

Because deletion of *Vegfr3* in mice leads to major cardiovascular defects and embryonic lethality (*Dumont et al., 1998*), we used an inducible knock out strategy in adult *Vegfr3*fl/fl mice (*Haiko et al., 2008*) that also contain an endothelium-specific, tamoxifen-inducible Cre (*Cdh5-Cre*ERT2) allele (*Wang et al., 2010*). *Cdh5 Cre*ERT2, *Vegfr3*fl/fl mice, referred as EC iΔR3, grow normally without any defect prior to tamoxifen injection. Two month old *Vegfr3*fl/fl (wild-type, WT) and EC iΔR3 mice were injected with tamoxifen and examined at 1, 2, 3 or 7 weeks. 1 week after tamoxifen injection, no VEGFR3 expression was visible in the thoracic aorta (*Figure 7B*) and in the ear skin lymphatics of EC iΔR3 mice (*Figure 7C*). 3 weeks after deletion of *Vegfr3*, the dermal lymphatic network in the skin was completely intact but vessel diameter was dramatically decreased (WT: 38 ± 5 μm and EC iΔR3: 22 ± 2 μm, n = 4, p < 0.001). We also observed a ~15% reduction of the diameter of the descending aorta (*Figure 7D,E*). No further change was observed when mice were examined at 7 weeks (*Figure 7E*), indicating that vessels remodeled and then stabilized. No change in body weight was observed 3 weeks after injection (28.4 g ± 2 for WT and 28.3 g ± 2.7 for EC iΔR3 mice). The curvature of the aortic arch was also reproducibly decreased after excision, an unexpected result that we have not further investigated.

To investigate the role of remodeling pathways, we stained longitudinal sections of the thoracic aorta for MMP9, a matrix metalloprotease involved in flow-dependent vascular remodeling (*Bond et al., 1998*; *Godin et al., 2000*; *Magid et al., 2003*). Following *Vegfr3* deletion, MMP9 in the thoracic aorta was highly elevated at 1 week but decreased to baseline at later times (*Figure 7F*). This observation strongly supports the notion that *Vegfr3* deletion induces inward remodeling of the thoracic aorta which is followed by stabilization. Increased MMP9 expression may be induced through NF-κB (*Sun et al., 2007*). We hypothesize that elevating the set point causes the endothelium to signal low shear, which induces inward remodeling. Together, these data support the concept that vessel lumen diameter is controlled by a VEGFR3-dependent shear stress set point.

## Discussion

Living organisms have developed an extensive repertoire of mechanisms to adapt to stresses and maintain homeostasis. For more than a century, investigators have observed effects suggesting that the blood flow controls vascular diameter (*Thoma, 1893*; *Langille and O'Donnell, 1986*; *Langille et al., 1989*; *Langille, 1996*), a mechanism that would optimize perfusion by adjusting vascular morphology in response to tissue demand. It has been hypothesized that, as for thermoregulation, there is an optimal value of flow which is maintained through feedback mechanisms to prevent deviation from this value. This is what we term the 'shear stress set point' theory (*Rodbard, 1975*). The current data show that HUVECs align in the direction of flow, inhibit NF-κB and activate Smads within a narrow range of fluid shear stress magnitudes. This range corresponds to the physiological flow within the umbilical vein estimated at around 8.4 to 12.5 dynes.cm$^{-2}$ ((*Kiserud and Rasmussen, 1998*; *Boito et al., 2002*; *Christensen et al., 2014*); shear stress = 8 × viscosity (velocity/diameter),

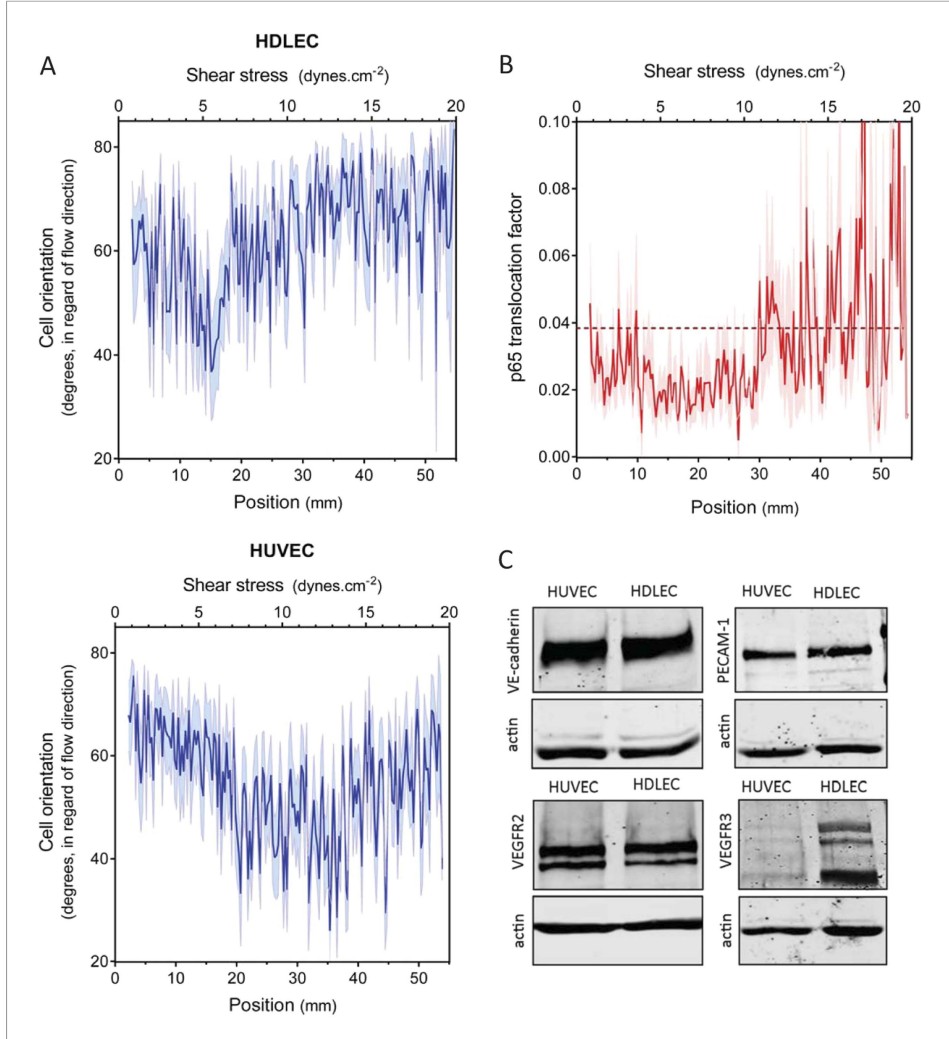

**Figure 3**. Set point in HUVECs vs lymphatic endothelial cells. (**A**) The average orientation of venous cell (HUVEC) or lymphatic cell (HDLEC) nuclei across the slide was measured as in *Figure 2A*. (n = 11, Mean ± SEM). The difference between HUVECs and HDLECs is statistically significant (ANOVA Two-way, p < 0.0001). (**B**) NF-κB activation: p65 nuclear translocation in HDLEC was measured either in no flow (dotted line: average) or after 16 hr of flow in the gradient chamber (n = 4, Mean ± SEM, ANOVA: F = 34.32, p < 0.0001). (**C**) Expression of VE-cadherin, PECAM-1, VEGFR2 and VEGFR3, proteins involved in the shear stress mechanotransduction through the junctional complex. Actin was used as a loading control.

The following figure supplement is available for figure 3:

**Figure supplement 1**. Representative pictures of HDLEC probed for DAPI and p65 at 5 and 20 dynes.cm$^{-2}$.

with viscosity = 0.06–0.09 poisse, velocity = 7.1 cm.s$^{-1}$ and diameter = 4.1 mm). These results imply that physiological flow inhibits inflammatory pathways and activates anti-inflammatory/stabilization pathways. By contrast, cells in low or high flow fail to align, activate NF-κB and suppress Smads. We propose that these responses are involved in the vessel remodeling that reestablishes optimal blood flow.

It is known that inflammation is a critical component of flow-dependent as well as other forms of vessel remodeling (*Silvestre et al., 2008*; *Schaper, 2009*; *Silvestre et al., 2013*). It has been recently demonstrated that inhibiting NF-κB impairs outward remodeling associated with increased shear stress as well as aneurysm formation (*Saito et al., 2013*). On the other hand, defective Smad1 signaling in the endothelium is associated with hereditary haemorrhagic telangiectasia (HHT), which is

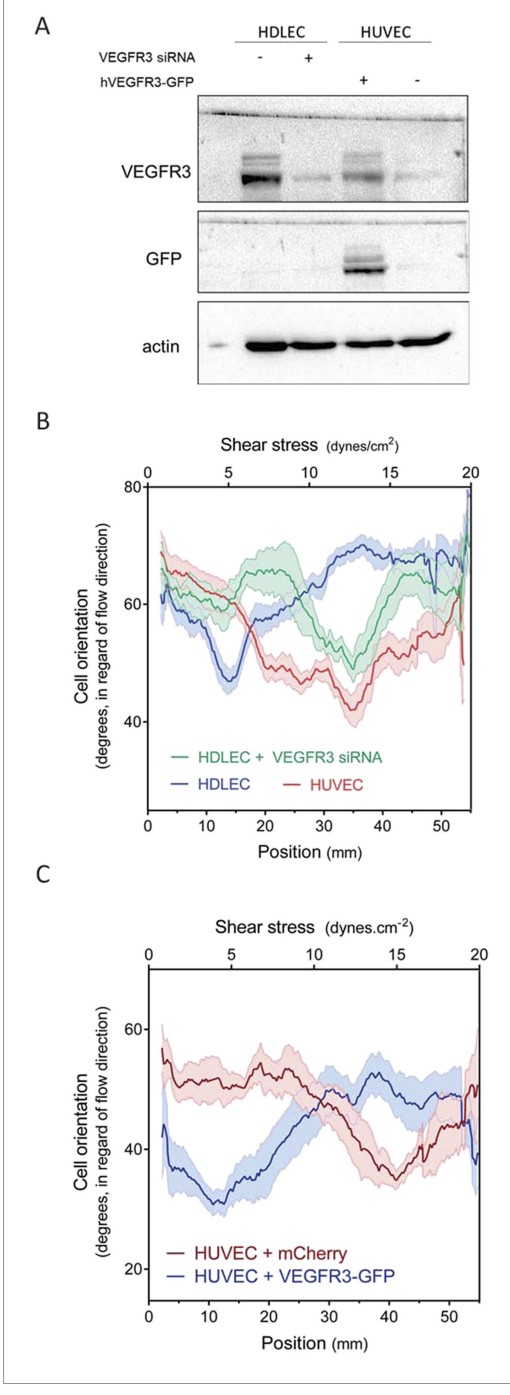

**Figure 4**. VEGFR3 expression controls the shear stress set point. (**A**) Western Blot of VEGFR3 and GFP in HDLECs with and without VEGFR3 siRNA (10 nM), and in HUVECs with and without adenoviral expression of hVEGFR3-GFP. Actin serves as a loading control. (**B**) Effect of VEGFR3 siRNA in HDLECs on set point. Cell alignment was assayed after shear stress for 16 hr (n = 6). Data were smoothed with a LOWESS fit to improve visualization (mean ± SEM; HDLEC vs HDLC + VEGFR3 siRNA: p = 0.004; HDLEC + VEGFR3 siRNA vs HUVEC: p = 0.45). (**C**) Effect of VEGFR3 over-expression on set

*Figure 4. continued on next page*

characterized by the development of unstable, arteriovenous malformations (*Dupuis-Girod et al., 2010*). Interestingly, these malformations are preceded by increased vascular lumen diameter, which occurs in a flow dependent manner (*Corti et al., 2011*). These observations, combined with ours, suggest that these two signaling pathways contribute to balanced control of the vessel caliber.

Fluid shear stress varies among different types of vessels, and to some extent even within the same vessel, suggesting that different cells must have different set points for shear stress, depending on their location. Relevant to our experiments, the shear stress in the human umbilical vein is estimated at around 8.4–12.5 dynes/cm$^{-2}$ whereas lymphatic vessels have highly pulsatile flow with peaks values at around 4–8 dynes.cm$^{-2}$ and averages that are much lower (*Dixon et al., 2006*). The shear stress set point model therefore predicts that these cell types will have different set points, which was borne out in our studies. Furthermore, we found that this difference can be largely accounted for by differences in VEGFR3 expression. This receptor, a close homolog of VEGFR2, is also activated in response to flow. Both expression levels in vivo (*Witmer et al., 2002*) and our functional experiments in vitro lead to the conclusion that high expression of VEGFR3 increases sensitivity to shear to give a low shear stress set point, while low expression of VEGFR3 is associated with higher set points. However, it is highly likely that other mechanotransducers or mediators influence set point values. While we did not observe any major difference in PECAM-1 and VE-cadherin expression between HDLEC and HUVEC, these two proteins can vary between different vascular beds (*Pusztaszeri et al., 2006*; *Herwig et al., 2008*), which might also affect the set point. We used HUVECs as a model for blood endothelial cells because they are readily available and their response to shear stress is well characterized. However, it has been recently showed that arterial and venous markers greatly diminish in culture (*Aranguren et al., 2013*), thus, whether they fully represent typical venous cells in vivo should be treated with caution. Comparing fresh primary cells from veins and arteries will be an interesting direction for future work. Mechanotransducers apart from the junctional complex are also likely to be important. There must also be pathways that distinguish high and low shear to initiate outward vs inward remodeling. Future work will be required to explore these

*Figure 4. Continued*

point. Alignment after 16 hr flow was assayed in HUVECs infected with adenovirus expressing mCherry or hVEGFR3-GFP as before. Data were smoothed with a LOWESS fit to improve visualization (n = 10, values are means ± SEM; HUVEC + mCherry vs HUVEC + VEGFR3-GFP: p < 0.0001).

The following figure supplements are available for figure 4:

**Figure supplement 1**. (**A**) Representative pictures of HUVEC cells expressing hVEGFR3-GFP (GFP signal displayed) after 16 hr of stimulation at 5 and 20 dynes.cm$^{-2}$.

**Figure supplement 2**. Non-smoothened data of the graphs displayed in *Figure 4B,C*.

pathways in more detail and their relevance to vascular remodeling.

The notion that vascular remodeling is governed by a shear stress set point, which is itself set by activation of various receptors and signaling pathways, may be relevant to a number of applications. Recovery from atherosclerotic luminal narrowing or myocardial infarction is thought to proceed in part via flow-dependent vessel remodeling (*Heil and Schaper, 2004*). Vascular graft adaptation also requires activation of signaling pathways activated by high shear stress to promote healing of the graft by preventing intimal proliferation (*Kohler et al., 1991*). Arteriovenous malformations are also thought to have a flow-dependent component (*Corti et al., 2011*). Thus, further understanding of the molecular sensors and downstream signaling pathways that control flow-dependent remodeling is relevant to a broad range of vascular dysfunction.

## Materials and methods

### Cell culture

Human Umbilical Vein Endothelial Cells (HUVECs) pooled from three different donors were obtained from the Yale Vascular Biology and Therapeutics program and cultured in M199 medium supplemented with 20% Fetal Bovine Serum, 50 µg.ml$^{-1}$ of Endothelial Cell growth Supplement (ECGS) prepared from bovine hypothalamus, 100 µg.ml$^{-1}$ heparin, 100 U.ml$^{-1}$ penicillin and 100 µg.ml$^{-1}$ streptomycin. They were used between passage 3 and 7. Human Dermal Lymphatic Endothelial Cells (HDLECs) were obtained from Lonza (Basel, Switzerland) and cultured in EGM-2 MV medium and used from passage 5 to 7. Cells were starved in M199 medium supplemented with 5% FBS and 100 U.ml$^{-1}$ penicillin and 100 µg.ml$^{-1}$ streptomycin for a minimum of 4 hr before further treatments.

### Shear stress

Cells were seeded on tissue culture plastic slides cut from 150 mm tissue culture dishes (Falcon), coated with 20 µg.ml$^{-1}$ fibronectin. Confluent cells were subjected to steady laminar shear stress in a modified parallel plate flow chamber (*Figure 1*) in which the gasket was a silicon sheet of either 0.8 or 1.6 mm height (Grace Bio-Labs, Bend, OR, #664172 and #664283) cut to generate a linear gradient of shear stress, calculated from (*Usami et al., 1993*). Flow was applied for 16 hr in starvation medium. Cells were then fixed with 4% formaldehyde in PBS for 10 min, permeabilized with 0.5% Triton x-100 in PBS for 10 min, blocked with Startingblock buffer (ThermoScientific) for 30 min at room temperature and probed overnight at 4°C with a primary antibody diluted in Startingblock buffer. Slides were stained with Hoechst 33342 to label nuclei, with rabbit anti-p65 antibody (Cell Signaling) to label NF-κB, and with rabbit anti-Smad1 antibody (Cell Signaling).

### Image analysis

Images were acquired with a Perkin Elmer spinning disk confocal microscope equipped with an automated stage which was used to take successive pictures along the chamber channel. Masks of the images were made using a combination of an adaptive histogram equalization algorithm with intensity and size thresholding. Cell orientation was calculated by taking the masks of the cell nuclei, fitting to an ellipse, and finding the angle between the flow direction and the major-axis of the ellipse. Nuclear translocation was computed by taking the mask of the nucleus and determining the integrated intensity of the transcription factor stain (Smad1 or p65) in the nucleus and in the whole cell. The 'translocation factor' (TF) was calculated by dividing the integrated intensity in the nucleus by the value for the whole cell. If the entire signal is localized to the nucleus, TF = 1, while if the entire signal is cytoplasmic, TF = 0.

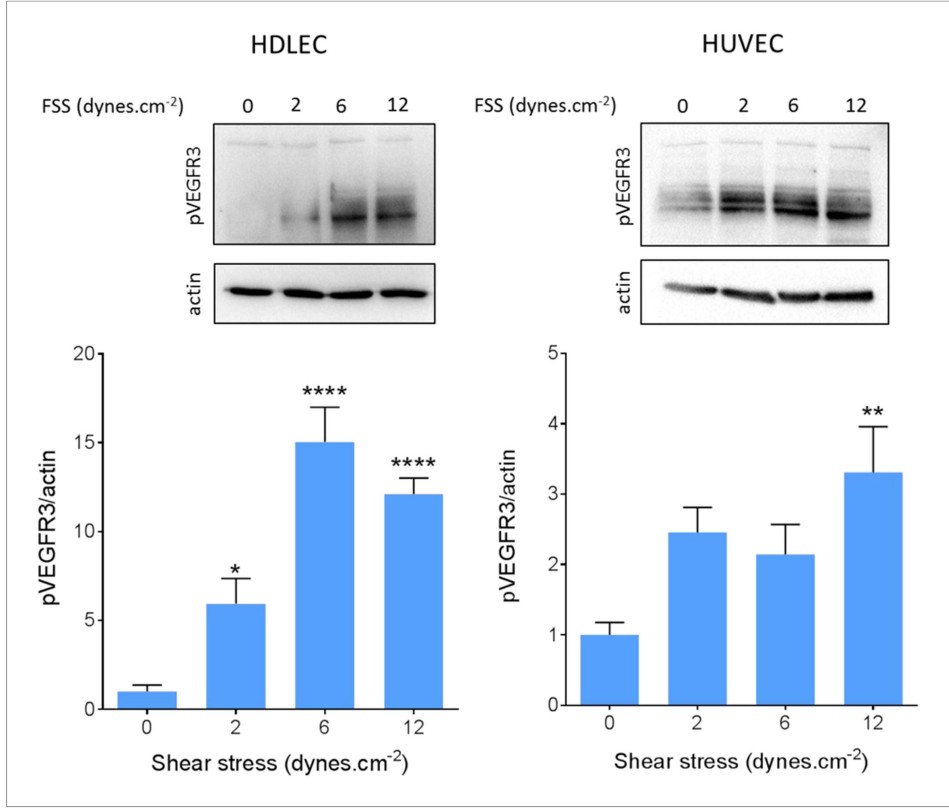

**Figure 5**. VEGFR3 activation by shear stress. HDLECs (left) and HUVECs (right) were stimulated for 15 min with shear stress at the indicated levels. VEGFR3 transactivation was assayed by phosphorylation on Y1230, detected by Western blotting with pY1230 antibody (n = 5 independent experiments; *: $p < 0.05$, **: $p < 0.01$, ****: $p < 0.0001$).

## siRNA transfection and adenoviral expression

Depletion of VEGFR3 was achieved by transfecting 10 nM siRNA (L-003138-00 OnTarget Smartpool Human FLT4, ThermoScientific) with Lipofectamine RNAi Max (Invitrogen), following the manufacturer's instructions. Transfection efficiency was assessed by Western-blot. Human VEGFR3-GFP was cloned in adenoviral (pAd) expression vector. Cells were infected with the virus in medium with polybrene (5 mg/ml) overnight and used 48 hr later.

## FACS

GFP expression in HUVEC or HUVEC infected with VEGFR3-GFP was assayed on a Stratedigm S1000EX (Stratedigm, San Jose, CA). Data were analyzed with the FlowJo software (TreeStar, Ashland, OR).

## Western blotting

Cells were washed with cold PBS and proteins extracted with Laemmli's buffer. Samples were run on 10 or 12% SDS-PAGE and transferred onto nitrocellulose membranes. The membranes were blocked with StartingBlock buffer (ThermoScientific) and probed with primary antibodies overnight at 4°C: VEGFR3 (R&D systems), phospho-VEGFR3 (Cell Applications), VEGFR2 (Cell Signaling), PECAM-1 (Abcam), VE-cadherin (Santa Cruz), GFP (Invitrogen) and actin (Santa Cruz). DyLight conjugated fluorescent secondary antibodies (680 nm and 800 nm, Thermoscientific) or HRP-conjugated antibodies were used to detect primary antibodies. Bands were detected and quantified with an Odyssey infrared imaging system for DyLight antibodies (Li-Cor) or a BioRad western blot imaging system (Bio Rad).

## Zebrafish

Zebrafish were grown and maintained according to protocols approved by the Yale University Animal Care. The *Tg(kdrl:mCherry; flt4:citrine)* was used (*Bussmann and Schulte-Merker, 2011*).

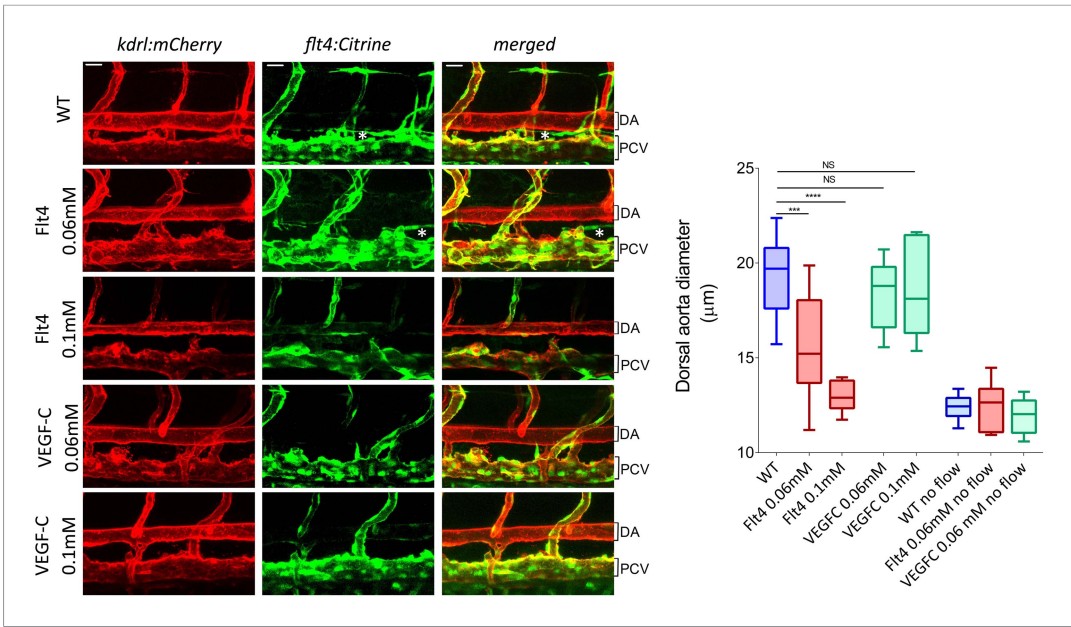

**Figure 6**. VEGFR3 (Flt4) controls blood vessel caliber in zebrafish. Representative pictures of the dorsal aorta (DA), posterior cardinal vein (PCV) and thoracic duct (white *) at 72 hr post-fertilization (hpf) in wild type zebrafish embryos or embryos injected with Flt4 (VEGFR3) morpholino at 0.06 or 0.1 mM, or with VEGF-C morpholino at 0.06 or 0.1 mM. The mCherry reporter driven by the KDR (VEGFR2) promoter (*kdrl:mCherry*) is depicted in red and the citrine reporter driven by the Flt4 promoter (*flt4:citrine*) is depicted in green. Scale = 20 μm and applies to all pictures. n = 6-15 fishes for each condition, whiskers represents the minimum and maximum data point (NS: non-significant, ***: p < 0.001 and ****: p < 0.0001, ANOVA).

The following figure supplement is available for figure 6:

**Figure supplement 1**. Representative pictures of the dorsal aorta (DA) and posterior cardinal vein (PCV) and developing thoracic duct (*) in wild type zebrafish embryos with a citrine reporter associated to Flt4 promoter (flt4: citrine) before and approximatively 2 hr after nifedipine treatment.

Morpholinos (*Nicoli et al., 2012*) were injected at the indicated concentrations and morphants were observed in a confocal microscope (SP5 Leica Microsystems). Images captured using Leica application suite software. Chemical treatment with nifedipine 40 μM was performed as previously described, 4 hr prior imaging (*Bussmann et al., 2011*).

## Mice

All animal experiments were approved by the Institutional Care and Use Committee of Yale University. The *Vegfr3::YFP* (*Calvo et al., 2011*), *Cdh5^CreERT2* (*Pitulescu et al., 2010*; *Wang et al., 2010*), *Vegfr3^flox/flox* (*Haiko et al., 2008*) mice were described previously. *Cdh5^CreERT2* mice were crossed with *Vegfr3^flox/flox* mice to generate endothelial-specific inducible *Vegfr3* mutant mice. 6–8 weeks old *Vegfr3^flox/flox* mice, with or without the Cre recombinase, were injected intra-peritoneally with 2 mg tamoxifen (TX; at 20 mg/ml in peanut oil (Sigma) with 10% Ethanol) once per day for 5 consecutive days (induction period). Mice were euthanized then fixed by perfusion with 3.7% formaldehyde 1, 2, 3 or 7 weeks after induction. Ear tissue was fixed overnight in 3.7% formaldehyde. The ear skin was removed, cleaned of connective tissue and cartilage, and permeabilized for 4 hr in permeabilization buffer (1% BSA, 1% NGS, 0.5% Tween in PBS). The skin was then incubated with antibody against LYVE-1 or VEGFR3, in 50% permeabilization buffer/50% PBS for 2 days at 4°C. After washing, the skin was incubated with secondary antibodies overnight 4°C in the same buffer. The skin was then flat mounted in Fluoromount G (Southern Biotech) and imaged with a Perkin Elmer spinning disk confocal microscope with a 20× objective. The aorta was removed, cleaned of all connective tissue, fixed overnight in 3.7% formaldehyde at 4°C and embedded in paraffin. Paraffin embedding and sectioning

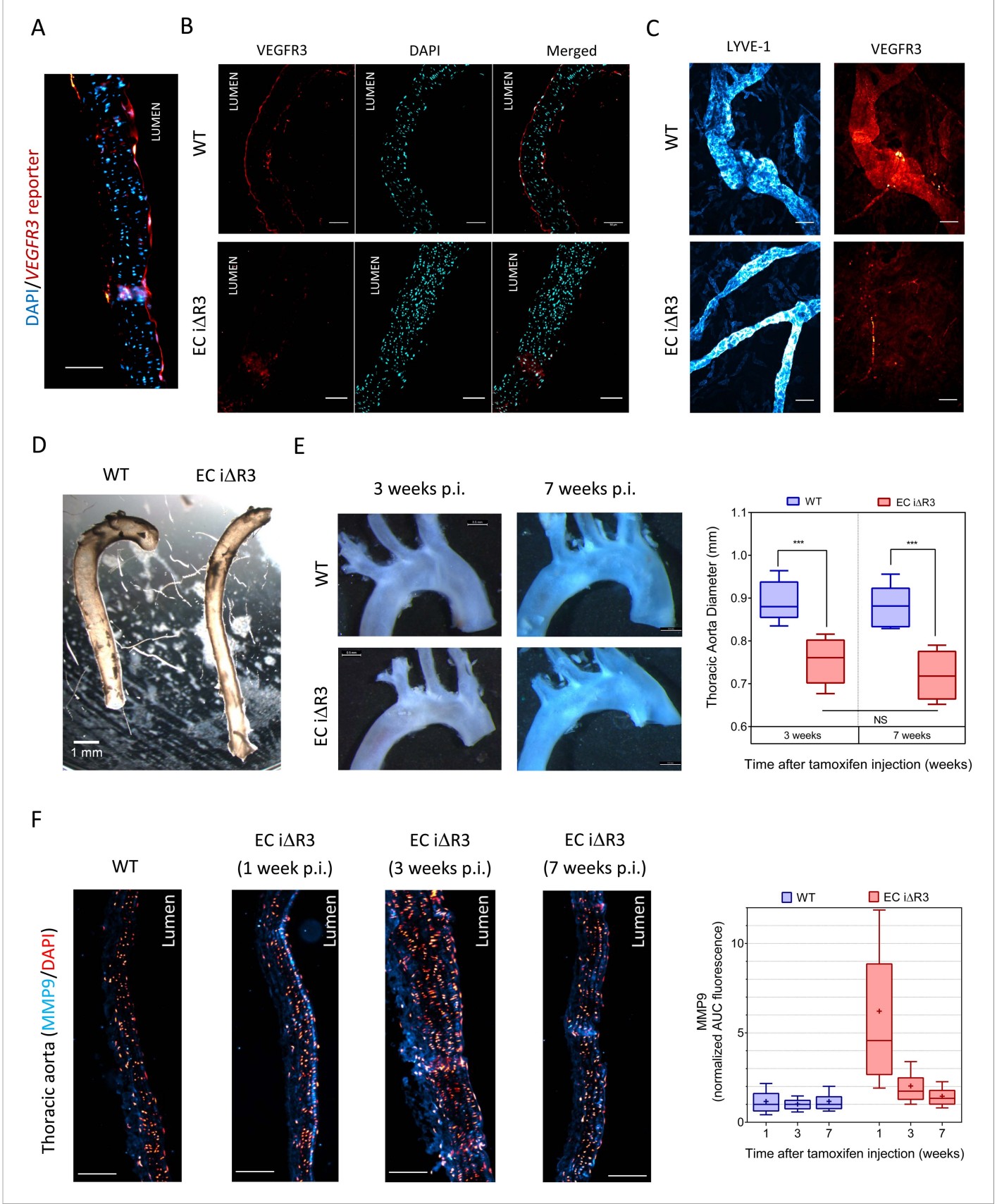

*Figure 7. Continued*

**Figure 7**. Transient vascular remodeling in *EC iΔR3* mice. (**A**) Longitudinal paraffin section of the thoracic aorta of a *Vegfr3::YFP* (VEGFR3 reporter) mouse. YFP was detected with an anti-GFP antibody. Scale bar = 50 µm. (**B**) VEGFR3 and DAPI staining of a longitudinal section of the thoracic aorta of *Vegfr3^{fl/fl}* (WT) or *EC iΔR3* mice, 3 weeks after Tx injection. Scale bar: 50 µm. (**C**) Lyve1 and VEGFR3 staining of the lymphatic network in ear skin from *Vegfr3^{fl/fl}* (WT) or *EC iΔR3* mice. Pictures were taken 1 week after Tx injection. Scale bar = 50 µm. (**D**) Aorta from oil injected-*EC iΔR3* (WT) or Tx-injected *EC iΔR3* mice, 2 weeks after Tx injection. Scale bar = 1 mm. (**E**) Diameters (graph on right) were measured in thoracic aortas (images on left) from *Vegfr3^{fl/fl}* (WT) or *EC iΔR3* mice, 3 weeks (WT: n = 8 and *EC iΔR3*: n = 7) and 7 weeks (WT: n = 6 and *EC iΔR3*: n = 5) after Tx treament (whiskers indicate the minimum and maximum data point, ***: p < 0.001, ANOVA). The measurement was performed right after the curvature, 1 mm below the subclavian artery bifurcation. (**F**) Longitudinal paraffin sections of the thoracic aorta from *Vegfr3^{fl/fl}* (WT) or *EC iΔR3* mice or *EC iΔR3* mice, probed for MMP9 (blue) and nuclei (red) after injection of Tx for the indicated time (WT is 1 week post-injection). Distribution of the area under the curve of MMP9 fluorescence from the media is plotted on the left (n ≥ 3 mice for each condition, whiskers are 10–90%, cross is the arithmetic mean).

The following figure supplement is available for figure 7:

**Figure supplement 1**. VEGFR3 and DAPI staining of a longitudinal section different portions of the aorta. Scale bar: 50 µm.

were performed by the Yale Pathology department, in the Tissue Microarray facility. Aortas were cut longitudinally, paraffin was removed in xylene baths and sections progressively rehydrated before antigen retrieval for 30 min at 95°C in citrate buffer (10 mM sodium citrate, 0.05% Tween, pH = 6). Sections were blocked for 30 min in StartingBlock blocking buffer (ThermoScientific) and probed either with anti-MMP9 antibody (Abcam, 1/400), anti-VEGFR3 antibody (R&D) or anti-GFP antibody (Invitrogen, 1/400). Slides were then washed 3× in PBS-Tween and once in PBS, then incubated with donkey-anti rabbit AlexaFluor 647 secondary antibody (Molecular Probes, 1 hr at RT, 1/500). Slides were washed 3× in PBS-Tween and once in PBS, then mounted in Fluoromount G (Southern Biotech). Slides were imaged with a Nikon Eclipse 80i epifluorescence microscope. Image analysis of MMP9 staining was performed by measuring the area under the curve of the fluorescence signal coming from the media in 4 different 20× pictures for each individual aorta. The fluorescence profile was obtained with MeasureEndo, an ImageJ macro.

## Statistics

Values indicated in the text are mean ± SD. At least three independent experiments were performed for each condition. Statistical tests were performed by using either analysis of variance tests (ANOVA) or unpaired Student's *t*-tests. The ANOVA test performed on *Figures 2 and 3* tested the null hypothesis that shear stress magnitude does not have an effect on either cell orientation or p65 and Smad1 nuclear translocation.

## Acknowledgements

We thank Dr Yingdi Wang for her help with the ear skin dissection and lymphatic vessels labeling, David D Simon for his help with the gradient chamber design and Dr Marleen Ansems for the acquisition of the FACS data. NB was supported by BAEF fellowship, WBI. World excellence scholarship and American Heart Association postdoctoral fellowship (14POST19020010). The work was supported by USPHS grant PO1 HL107205 to MAS.

## Additional information

### Funding

| Funder | Grant reference number | Author |
|---|---|---|
| U.S. Public Health Service | P01 HL107205 | Martin A Schwartz |
| American Heart Association (AHA) | 14POST19020010 | Nicolas Baeyens |

The funders had no role in study design, data collection and interpretation, or the decision to submit the work for publication.

## Author contributions

NB, Conception and design, Acquisition of data, Analysis and interpretation of data, Drafting or revising the article; SN, Acquisition of data, Analysis and interpretation of data; BGC, Helped with experimental design, Analysis and interpretation of data; TDR, KVD, COM, Development of analytical tools; JH, Acquisition of data, Contributed unpublished essential data or reagents; HML, Design of the flow chamber; AE, J-LT, Provided critical ideas about the use of the VEGFR3 endothelial inducible KO mice, Contributed unpublished essential data or reagents; JDH, Provided the original idea, Contributed unpublished essential data or reagents; MAS, Provided the original idea, Conception and design, Analysis and interpretation of data, Drafting or revising the article

## Ethics

Animal experimentation: All animal experiments were performed in accordance with the recommendations in the Guide for the Care and Use of Laboratory Animals of the National Institutes of Health and approved by the Institutional Care and Use Committee of Yale University (protocol #11406).

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
