## [Decision Letter]

Thank you for sending your work entitled “Vascular remodeling is governed by a VEGFR3-dependent fluid shear stress set point” for consideration at *eLife*. Your article has been favorably evaluated by Fiona Watt (Senior editor) and three reviewers.

The Senior editor and the reviewers discussed their comments before we reached this decision, and the Senior editor has assembled the following comments to help you prepare a revised submission. All the reviewers considered your work to be interesting and important. Although there is a long list of comments for you to address, many can be dealt with in the text and by strengthening the Discussion.

However, two critical issues are as follows:

First, the cited unpublished results (Coon et al.) that VEGFR3 is a part of the mechanosensory complex consisting of PECAM–1, VEC and VEGFR2 should be presented in this manuscript. It is impossible to accept the authors' statements without reviewing this evidence.

Second, there does not seem to be any detectable Flt4 signal in the zebrafish aorta. Previously published results indicate that Flt4 is expressed in the PCV, not aorta, and in segmental arteries, and Vegfc is expressed in the zebrafish aorta (Covassin et al., PNAS, 2006; Siekman and Lawson, Nature, 2007). How could Flt4 MO have any effect on aorta width in this case? High MO concentrations could have off-target effects in aorta development, or on blood flow. In fact, recent results have indicated that few phenotypes based on morpholinos are similar to those based on corresponding gene-edited deletions, casting doubt on experiments that employ morpholinos (see Schulte-Merker and Stainier, 2014, Development, 3103-3104).

Specific points:

1) Flow chamber:

a Please explain what is the actual flow profile. It is unclear if the linear dependence shown in Figure 1 was obtained experimentally or calculated using laminar flow models.

b. Are there any downstream effects of the cells exposed to low shear to those exposed to higher shear?

c. Please clarify how reproducible are data shown in Figure 2 (all three panels). Could statistical differences be shown?

2) It is stated that HUVECs aligned between 8 and 20 dynes/cm^2^ (Figures 2 and 3), lymphatic endothelial cells aligned between 4 and 6 dynes/cm^2^, (Figure 3), and the minimum NF-kB translocation shifted to 4 and 10 dynes/cm^2^ (Figure 3). This is not obvious from the data shown. It would be helpful to clarify how these ranges were determined, and if they correspond to statistically significant differences relatively to the other regions. In Figure 2 representative pictures of cell alignment (junctional staining or bright field images or at least the one in supplemental figure) should be included for 3 values of flow (under 10, between 10 and 20, above 20 dynes/cm^2^).

3) Figure 4: Again, it would be instructive to know what is the reproducibility of these data (which look like single data sets), and important to determine which differences are significant.

4) The zebrafish loss-of-function experiments are not temporally controlled and flt4 will have additional roles that are difficult to separate from the alleged mechanosensory role. Therefore the authors should avoid overstating these results in saying that they show that ligand independent activation of flt4 is involved in this process. The authors do not provide any data on flt4 activation at all. So it remains unclear how, mechanistically, VEGFR3 is involved.

5) The authors should include representative images of cell alignment, p65 and Smad1 under different shear stress conditions. It should be noted in the text that the quantification shown in the figure uses nuclear anisotropy as a proxy for cell alignment. This may well be justifiable but needs to be made clear. The authors assess alignment and NFκB–SMAD translocation at 16 hr for both HUVECs and HDLEC. Could these cells have also a different time-response to shear stress to alignment and NFκB–SMAD activation?

6) Is MMP9 part of the set-point by affecting “stiffness”, or just a downstream effector involved in remodelling? If not possible to tie down experimentally, this should at least be discussed. Are p65 and Smad1 affected in vivo? This should be tested to better link the parts of the manuscript.

7) Regarding the quantification program for the translocation factor, how do the authors differentiate between cells with very low level of NFkB everywhere (cytoplasm and nucleus with the same staining intensity) and cells with very high level of NFkB everywhere? Biologically these two situations are different because NFκB increases in the nucleus but the ratio would give the same value. In other words, are total levels affected by shear levels and is this relevant? Can it be excluded? Same comment for Smad1.

8) The authors mention in the text that more than 90% of cells expressed VEGFR3-GFP. Based on the images provided in Figure 4—figure supplement 1, we can see approximately half of the cell GFP+ (or it is not a confluent monolayer). The authors should assess the transduction efficiency with a more accurate method.

9) Please comment on what happens to the thoracic duct in nifedipine treated fish.

10) The authors nicely show that VEGFR3 is expressed in EC of thoracic Aorta. As the inner part of the curvature of the aorta is exposed to lower shear stress, do the authors see any differential expression of VEGFR3 in this area? Is there any regulation of VEGFR3 expression or phosphorylation by the shear stress level in the in vitro chamber?

11) The authors observe a strange effect on aorta curvature in EC iDR3 mice. Could it be linked with the increased production of MMPs potentially softening the surrounding matrix. Could the authors comment on this?

12) How is MMP9 expression modulated in the flow chamber model?

13) Do the authors have any explanation on how VEGFR3 expression regulates MMP9 expression?

14) How are p65 and SMAD modified in the aorta of EC iDR3 mice or in the different morpholino fish? Is the alignment of EC modified?

15) Could it be that VEGFR3 deletion has a short time effect on vessel diameter by modulating the vasomotor tone in addition to the remodelling effect of MMPs production?

16) The YFP fluorescent reporter seems to be based on a large recombinant genomic clone in an unknown chromosomal location, where it can be influenced by the chromosomal context. Does the YFP expression reflect reliably the expression of VEGFR3 in the adult aorta? What is the half-life of the YFP used in Figure 6? The YFP expression appears “patchy” in Figure 6; does this mean that ECs in the aorta expresses variable amounts of VEGFR3? The authors should obtain convincing data showing that the VEGFR3 protein is indeed expressed at a significant level in the aortic endothelial cells in vivo.

17) The authors use HUVECs in their in vitro studies. These cells do not correspond to the endothelial cells in the mouse or zebrafish aorta. Although the flow conditions in umbilical vein may be similar to those in an artery, venous endothelial cells express higher levels of VEGFR3 than arterial endothelial cells (for example, see Hogan et al., Development, 2009). Somehow, based on in vitro analysis, the shear stress “setpoints” are handled rather indiscriminately, considering that arteries and veins as well as primary lymph vessels and collecting vessels also have obvious differences in shear stress in vivo. It would be useful and relevant to perform the gradient flow chamber experiments using freshly isolated arterial endothelial cells, or even better, aortic endothelial cells. If the authors are not willing to do this, then at least the limitation of using the cell lines should be discussed.

18) The completeness of recombination achieved in the mouse aorta using the *Cdh5-Cre* line produced by Wang et al. is not reported. How does the dermal lymphatic data shown in Figure 6 connect with the rest of the manuscript? It certainly does not provide surrogate evidence for VEGFR3 deletion, and should be replaced by the assessment of VEGFR3 expression levels in mouse aortas before and after *Cre*-mediated deletion by IHC, WB and qPCR.

[Editors' note: further revisions were requested prior to acceptance, as described below.]

Thank you for resubmitting your work entitled “Vascular remodeling is governed by a VEGFR3-dependent fluid shear stress set point” for further consideration at *eLife*. Your revised article has been evaluated by Fiona Watt (Senior editor) and the original reviewers. The manuscript has been improved but there are some remaining issues that need to be addressed before acceptance.

Reviewer #1 requests a number of specific clarifications, as listed below. In addition Reviewers #2 and #3 flag up the problem of how to avoid duplication of data in your two manuscripts, while nevertheless giving *eLife* readers sufficient data to enable them to evaluate your findings without having to refer to the Coon et al paper. We have copied verbatim some of Reviewer #3's comments on this issue.

We believe that you can address the reviewers' concerns without further experimentation. It is harder to see a solution to the problem of overlapping findings in the two papers and would appreciate your comments.

Reviewer #1:

1) Flow chamber:

a) Critique: Please explain what is the actual flow profile It is unclear if the linear dependence shown in Figure 1 was obtained experimentally or calculated using laminar flow models.

Response: “The linear dependence of the flow profile was calculated using flow models from the original manuscript describing this chamber (Usami, Chen et al. 1993). To confirm the results obtained with the gradient chamber, we performed additional experiments with a conventional parallel plate chamber under uniform shear stress of 2, 12 or 50 dynes.cm^-2^.”

There is still no clear and direct validation of model predictions. The authors should either validate or remove the model predictions.

b) Critique: Are there any downstream effects of the cells exposed to low shear to those exposed to higher shear?

Response: “The results of these experiments (Figure 2–figure supplement) support the results obtained with the gradient chamber, ruling out downstream effect from cells exposed to low shear on the responses measured at higher shear stress.”

Again, it is not obvious how the responses to uniform shear at three different levels rule out the downstream effects from cells exposed to low shear to cells exposed to higher shear.

c) Critique: Please clarify how reproducible are data shown in Figure 2 (all three panels). Could statistical differences be shown?

Response: “Lastly, the statistical significance of the results obtained with the gradient chamber has been tested with an ANOVA test and the result of the test for each variable is now included in the legends of the Figure 2 and Figure 2—figure supplement 1.”

It remains unclear how the statistics was done and what it tells us about the reproducibility of the data. I see the response unrelated to the question.

Also, the responses to points 12 and 13 about the MMP 9 expression are vague. The authors state that they could not observe any change in the MMP 9 intensity in HUVECs in the gradient chamber and conclude that “MMP9 analysis in vitro does not appear to be a fruitful line of investigation”, whereas there is a body of literature reporting in vitro studies of MMP 9 expression in various cell types. Similarly, they do not offer an explanation on how VEGFR3 expression regulates MMP9 expression.

Reviewer #3:

The Coon et al. manuscript that has apparently been sent to another journal presents some complications:

1) There is overlap as in both manuscripts induction of VEGFR3 phosphorylation by FSS is now presented as a novel finding. The new Figure 5 in the revised *eLife* manuscript is not of the necessary standard as it fails to show total VEGFR3 protein in the lysates, only P-VEGFR3 is displayed. The total VEGFR3 protein is analysed in a similar Figure 4 of the other manuscript, but one cannot ask the *eLife* readers to consult another paper for such basic control.

2) The *eLife* manuscript avoids clearly mentioning the VEC;VEGFR2 complex, whereas the other manuscript claims VEGFR2;VEGFR3;VEC mechanosensory complex is involved, and speculates that a critical component may actually be a phosphatase.

3) Whereas the *eLife* manuscript emphasizes VEGFR3 alone as the FSS set point, the other manuscript says in the abstract that “VEGFR2 and VEGFR3 signal redundantly downstream of VE-cadherin”.

4) Both manuscripts also show evidence of “significant” VEGFR3 expression in aortic endothelium. It is difficult to know if this amount is biologically significant.

5) The *eLife* abstract says: “VEGFR3 modulates arterial lumen diameter consistent with flow-dependent remodeling” (it should probably be “aortic” rather than “arterial” here, as other arteriae were not studied). The abstract of the other manuscript has a similar result: “VEGFR3 expression is observed in the aortic endothelium where it contributes to flow responses in vivo”.

---

## [Author Response]

*First, the cited unpublished results (Coon et al.) that VEGFR3 is a part of the mechanosensory complex consisting of PECAM–1, VEC and VEGFR2 should be presented in this manuscript. It is impossible to accept the authors' statements without reviewing this evidence*.

The main point that is relevant to the current study is that VEGFR3 contributes to flow signaling similarly to VEGFR2. The mechanism by which it is activated by flow is a separate issue that is better left for the other paper where it is studied in detail. We have therefore added new data demonstrating activation of VEGFR3 by flow in both HUVECs and lymphatic ECs (Figure 5) and provide the unpublished manuscript for the reviewers’ inspection.

*Second, there does not seem to be any detectable Flt4 signal in the zebrafish aorta. Previously published results indicate that Flt4 is expressed in the PCV, not aorta, and in segmental arteries, and VEGFC is expressed in the zebrafish aorta (Covassin et al., PNAS, 2006; Siekman and Lawson, Nature, 2007). How could Flt4 MO have any effect on aorta width in this case? High MO concentrations could have off-target effects in aorta development, or on blood flow. In fact, recent results have indicated that few phenotypes based on morpholinos are similar to those based on corresponding gene-edited deletions, casting doubt on experiments that employ morpholinos (see Schulte-Merker and Stainier, 2014, Development, 3103-3104)*.

We have now included a full-size, detailed picture of the Flt4 reporter embryo (Figure 6—figure supplement 1), where we observe a gradient of Flt4 transcription: high expression in the developing thoracic duct, lower in the posterior cardinal vein and lower but still readily detectable in the dorsal aorta. This observation is very much in accordance with our paradigm that expression of VEGFR3 inversely correlates with shear stress magnitude in vasculature. We also would like to stress that these former studies by Nathan Lawson’s laboratory were performed at 24 hpf, during the initial phases of vasculature development, while our observation is done at 72 hpf, when the vasculature is more mature.

Concerning the potential off-target effect of the morpholinos, the concentration dependent effect on dorsal aorta diameter without affecting the whole vasculature of the embryo or its viability, supports the specificity of the effect. More critically, the kinase dead *expando* mutant for Flt4 (Hogan, Herpers et al., 2009) yielded similar effects on the diameter of the dorsal aorta, though these were not noted by the authors (Kwon, Fukuhara et al., 2013). This effect is not observable with a VEGFC morpholino, in accordance with our own observations. These results, plus the similar observation in mice, strongly support the conclusion that the decrease of the dorsal aorta diameter is related to the decreased levels of Flt4 rather than an off-target effect. All of these issues are now discussed in the manuscript.

*Specific points*:

*1) Flow chamber*:

*a. Please explain what is the actual flow profile. It is unclear if the linear dependence shown in*
Figure 1
*was obtained experimentally or calculated using laminar flow models*.

b. Are there any downstream effects of the cells exposed to low shear to those exposed to higher shear?

*c. Please clarify how reproducible are data shown in*
Figure 2
*(all three panels). Could statistical differences be shown?*

The linear dependence of the flow profile was calculated using flow models from the original manuscript describing this chamber (Usami, Chen et al., 1993). To confirm the results obtained with the gradient chamber, we performed additional experiments with a conventional parallel plate chamber under uniform shear stress of 2, 12 or 50 dynes.cm^-2^. The results of these experiments (Figure 2—figure supplement 1) support the results obtained with the gradient chamber, ruling out downstream effect from cells exposed to low shear on the responses measured at higher shear stress. Lastly, the statistical significance of the results obtained with the gradient chamber has been tested with an ANOVA test and the result of the test for each variable is now included in the legends of the Figure 2 and Figure 2—figure supplement 1.

*2) It is stated that HUVECs aligned between 8 and 20 dynes/cm*^*2*^
*(*Figures 2 and 3*), lymphatic endothelial cells aligned between 4 and 6 dynes/cm*^*2*^*, (*Figure 3*), and the minimum NF-κB translocation shifted to 4 and 10 dynes/cm*^*2*^
*(*Figure 3*). This is not obvious from the data shown. It would be helpful to clarify how these ranges were determined, and if they correspond to statistically significant differences relatively to the other regions. In*
Figure 2
*representative pictures of cell alignment (junctional staining or bright field images or at least the one in supplemental figure) should be included for 3 values of flow (under 10, between 10 and 20, above 20 dynes/cm*^*2*^*)*.

The set point ranges given in the test were determined by visual inspection of the data and, as indicated in the text, are approximate. As requested, representative pictures for different flow magnitudes have been included, including phalloidin staining to highlight cell alignment. Statistics were determined with an ANOVA test and the result of the test for each variable is included in the legend of the Figure 2, Figure 2—figure supplement 1 and Figure 3.

*3)*
Figure 4*: Again, it would be instructive to know what is the reproducibility of these data (which look like single data sets), and important to determine which differences are significant*.

*4) The zebrafish loss-of-function experiments are not temporally controlled and flt4 will have additional roles that are difficult to separate from the alleged mechanosensory role. Therefore the authors should avoid overstating these results in saying that they show that ligand independent activation of flt4 is involved in this process. The authors do not provide any data on flt4 activation at all. So it remains unclear how, mechanistically, VEGFR3 is involved*.

We performed an ANOVA test to compare the different situations and the results are now included in the legend for Figure 4. We also included the raw data in Figure 4—figure supplement 2.

5) The authors should include representative images of cell alignment, p65 and Smad1 under different shear stress conditions. It should be noted in the text that the quantification shown in the figure uses nuclear anisotropy as a proxy for cell alignment. This may well be justifiable but needs to be made clear. The authors assess alignment and NFkB–SMAD translocation at 16h for both HUVECs and HDLEC. Could these cells have also a different time-response to shear stress to alignment and NFkB–SMAD activation?

We have added representative pictures and have highlighted that cell alignment was quantified by measuring nuclei orientation. We previously validated that nuclear alignment is a suitable metric that is highly correlated with cell alignment and that reference is now cited (Baeyens, Mulligan-Kehoe et al., 2014). Regarding the time dependence of the responses, our opinion is that this may not be relevant to the set point theory: lymphatic cells can align at a magnitude where HUVEC cannot, after 16 hours of flow, indicating clearly that the shear sensing mechanisms are optimized for lower values of shear stress in lymphatic cells.

*6) Is MMP9 part of the set-point by affecting “stiffness”, or just a downstream effector involved in remodelling? If not possible to tie down experimentally, this should at least be discussed. Are p65 and Smad1 affected in vivo? This should be tested to better link the parts of the manuscript*.

We examined MMP9 as a well-known downstream effector in arterial remodeling primarily to assess the time course of remodeling. The key point from the results is that following MMP9 expression after VEGFR3 deletion, it then resolves, indicating that vessels have reached a new steady state, thus, are not undergoing regression or some other continuing process. Its expression is thought to be controlled in part by NFkB but also by other factors whose identification is not important here. We have been unable to reliably quantify p65 and SMAD nuclear translocation in longitudinal sections of the thoracic aorta.

*7) Regarding the quantification program for the translocation factor, how do the authors differentiate between cells with very low level of NFkB everywhere (cytoplasm and nucleus with the same staining intensity) and cells with very high level of NFkB everywhere? Biologically these two situations are different because NFkB increases in the nucleus but the ratio would give the same value. In other words, are total levels affected by shear levels and is this relevant? Can it be excluded? Same comment for Smad1*.

While this is in principle a valid point, there are in fact no discernable changes in staining intensity but rather in staining localization (Figure 2—figure supplement 1). Western blotting also shows little change in total p65 expression under flow (Orr, Sanders et al., 2005). This is why we developed the high-throughput quantification method, allowing us to assess this very specific feature of nuclear translocation, which is a well-established feature of NFκB and Smad1 activation.

*8) The authors mention in the text that more than 90% of cells expressed VEGFR3-GFP. Based on the images provided in*
Figure 4—figure supplement 1*, we can see approximately half of the cell GFP+ (or it is not a confluent monolayer). The authors should assess the transduction efficiency with a more accurate method*.

While this is in principle a valid point, there are in fact no discernable changes in staining intensity but rather in staining localization (Figure 2—figure supplement 1). Western blotting also shows little change in total p65 expression under flow (Orr, Sanders et al. 2005). This is why we developed the high-throughput quantification method, allowing us to assess this very specific feature of nuclear translocation, which is a well-established feature of NFκ B and Smad1 activation.

9) Please comment on what happens to the thoracic duct in nifedipine treated fish.

Although the thoracic duct is still developing at 3dpf (Yaniv, Isogai et al., 2006), we observed very clearly that it still developed in the presence of nifedipine (Figure 6—figure supplement 1).

10) The authors nicely show that VEGFR3 is expressed in EC of thoracic Aorta. As the inner part of the curvature of the aorta is exposed to lower shear stress, do the authors see any differential expression of VEGFR3 in this area? Is there any regulation of VEGFR3 expression or phosphorylation by the shear stress level in the in vitro chamber?

We do observe higher VEGFR3 expression in the inner curvature and some portions of the carotids, and lower expression on the outer curvature of the aorta (Figure 7—figure supplement 1), suggesting that VEGFR3 expression is higher in areas of low shear stress. However, within each vessel, expression appears nonuniform, suggesting local regulation. We also have in vitro data suggesting that VEGFR3 is slightly reduced under flow mainly due to internalization and degradation. The factors that govern VEGFR3 expression in arterial endothelial cells therefore appear to be complex and represent a distinct question separate from the one we address in this manuscript. We feel this issue is best left for further work.

*11) The authors observe a strange effect on aorta curvature in EC iDR3 mice*. *Could it be linked with the increased production of MMPs potentially softening the surrounding matrix. Could the authors comment on this?*

Yes, it is plausible that asymmetric remodeling of the aorta involving MMP9 resulted in the change in aorta curvature. Such asymmetric remodeling may be the consequence of a non-uniform VEGFR3 expression along the aorta. Or it could be a feature of the complex mechanics of the ascending aorta, which moves axially during each cardiac cycle and would be predicted to differentially load the inner and outer curvature (Kassab, 2006). We thought it appropriate to make a note of the observation but feel that anything we could say would be largely speculative as well as being unrelated to the main point of the paper and to any known remodeling process in vivo.

12) How is MMP9 expression modulated in the flow chamber model?

We stained for MMP9 in HUVECs in the gradient chamber but failed to observe any consistent change in intensity. However, MMP9 is a determinant of 3-dimensional tissue remodeling in vivo; ECs on a flat coverslip without a smooth muscle layer cannot undergo any sort of comparable process and may not express or capture secreted MMP9 in the same way. MMP9 analysis in vitro does not appear to be a fruitful line of investigation.

13) Do the authors have any explanation on how VEGFR3 expression regulates MMP9 expression?

Our current explanation is that reduction of VEGFR3 expression induces a change in the set point of the endothelial cells, which results in altered flow signaling, which induces remodeling. In essence, VEGFR3 deletion results in ECs sensing that shear levels are too low, which induces inward remodeling to increase shear. MMP9 may be a fairly indirect downstream consequence of these events.

14) How are p65 and SMAD modified in the aorta of EC iDR3 mice or in the different morpholino fish? Is the alignment of EC modified?

We tried to measure nuclear translocation in longitudinal sections of the aorta but have been unable to reliably quantify the effects in these sections. Expression levels of Smad1 and p65 seemed unaffected. Also, it is impossible, to our knowledge, to assess properly endothelial cell alignment in zebrafish embryos or in longitudinal sections of the aorta.

15) Could it be that VEGFR3 deletion has a short time effect on vessel diameter by modulating the vasomotor tone in addition to the remodelling effect of MMPs production?

Vasomotor tone is currently thought to be a critical component of vascular remodeling. The idea is that high or low shear induce, respectively, vasorelaxation or constriction, which over long times is entrained by tissue remodeling (Humphrey, Dufresne et al., 2014). It is also well established that production of nitric oxide in response to flow is mediated in part by the PECAM/VE-cadherin-VEGFR2 pathway (Jin, Ueba et al., 2003; Fleming, Fisslthaler et al., 2005). Thus, a similar role for VEGFR3, which leads to vessel remodeling, is quite reasonable.

*16) The YFP fluorescent reporter seems to be based on a large recombinant genomic clone in an unknown chromosomal location, where it can be influenced by the chromosomal context. Does the YFP expression reflect reliably the expression of VEGFR3 in the adult aorta? What is the half-life of the YFP used in*
Figure 6*? The YFP expression appears “patchy”in*
Figure 6*; does this mean that ECs in the aorta expresses variable amounts of VEGFR3? The authors should obtain convincing data showing that the VEGFR3 protein is indeed expressed at a significant level in the aortic endothelial cells in vivo*.

To address this problem, we performed an immunohistological staining for VEGFR3 in the thoracic aorta and the ear skin (Figure 7). This staining matches our observation with the YFP fluorescent reporter.

*17) The authors use HUVECs in their in vitro studies. These cells do not correspond to the endothelial cells in the mouse or zebrafish aorta. Although the flow conditions in umbilical vein may be similar to those in an artery, venous endothelial cells express higher levels of VEGFR3 than arterial endothelial cells (for example, see Hogan et al., Development, 2009). Somehow, based on in vitro analysis, the shear stress “setpoints” are handled rather indiscriminately, considering that arteries and veins as well as primary lymph vessels and collecting vessels also have obvious differences in shear stress in vivo. It would be useful and relevant to perform the gradient flow chamber experiments using freshly isolated arterial endothelial cells, or even better, aortic endothelial cells. If the authors are not willing to do this, then at least the limitation of using the cell lines should be discussed*.

Our fundamental reason for using HUVECs is that since most of the venous vs arterial gene expression signature is lost when ECs are cultured (Aranguren, Agirre et al., 2013), we felt it was better to use cells that are easy to obtain and use, and that respond well to flow. The feature we are studying appears to be conserved among different types of ECs, thus, HUVECs provide a suitable in vitro model. We tried freshly isolated mouse arterial ECs but they failed to show any alignment in response to flow. This is very likely a protocol problem but in practical terms is a significant issue. We now include a statement about the limitation of using these two cells lines and invite the reader to consider that different cells lines may have different set points.

*18) The completeness of recombination achieved in the mouse aorta using the* Cdh5-Cre *line produced by Wang et al. is not reported. How does the dermal lymphatic data shown in*
Figure 6
*connect with the rest of the manuscript? It certainly does not provide surrogate evidence for VEGFR3 deletion, and should be replaced by the assessment of VEGFR3 expression levels in mouse aortas before and after* Cre*-mediated deletion by IHC, WB and qPCR*.

As mentioned earlier, we have performed an immunohistological staining of VEGFR3 in the thoracic aorta and the skin ear in both WT and VEGFR3 iKO. Figure 7 shows that VEGFR3 expression is drastically reduced after recombination.

*[Editors' note: further revisions were requested prior to acceptance, as described below*.*]*

Reviewer #1:

*1) Flow chamber*:

*a) Critique: Please explain what is the actual flow profile It is unclear if the linear dependence shown in*
Figure 1
*was obtained experimentally or calculated using laminar flow models*.

*Response: “The linear dependence of the flow profile was calculated using flow models from the original manuscript describing this chamber (Usami, Chen et al. 1993). To confirm the results obtained with the gradient chamber, we performed additional experiments with a conventional parallel plate chamber under uniform shear stress of 2, 12 or 50 dynes.cm*^*-2*^.*”*

*There is still no clear and direct validation of model predictions. The authors should either validate or remove the model predictions*.

We have not carried out any additional validation of the flow profile per se beyond what was done in the original publication (58). This point is now clarified in the Methods section. What we did do was validate the biological conclusions from the gradient chamber by using standard flow chambers. The new data that we added fully supported those conclusions, which we feel is the main issue.

b) Critique: Are there any downstream effects of the cells exposed to low shear to those exposed to higher shear?

*Response: “The results of these experiments (*Figure 2—figure supplement 1*) support the results obtained with the gradient chamber, ruling out downstream effect from cells exposed to low shear on the responses measured at higher shear stress*.*”*

*Again, it is not obvious how the responses to uniform shear at three different levels rule out the downstream effects from cells exposed to low shear to cells exposed to higher shear*.

The reviewer was concerned that medium flowing from the region of low shear stress could influence the cells in the downstream region under high shear stress. We therefore carried out experiments where each slide was under uniform shear. The effects confirmed the results with the gradient chamber. In that setting, there is no medium flowing from low to high shear, thus, we believe that this addresses the issue. Please also note that our hydraulic circuit comprises a circulating volume of 150 ml of culture media flowing at 1.5 ml by second, which should quickly dilute any soluble factor produced by the cells. Lastly, we also carried out experiments with standard flow chambers where the medium flowed in the opposite direction (high shear stress chamber to low shear stress chamber); these gave identical results.

*c) Critique: Please clarify how reproducible are data shown in*
Figure 2
*(all three panels). Could statistical differences be shown?*

*Response: “Lastly, the statistical significance of the results obtained with the gradient chamber has been tested with an ANOVA test and the result of the test for each variable is now included in the legends of the*
Figure 2
*and*
Figure 2—figure supplement 1.*”*

*It remains unclear how the statistics was done and what it tells us about the reproducibility of the data. I see the response unrelated to the question*.

We now provide additional clarifications about the way the statistics were done. The one way ANOVA test performed on the data in Figure 2 tested the null hypothesis that shear stress magnitude has no effect on endothelial alignment, p65 and Smad1 nuclear translocation. This analysis showed very clearly that it is not the case. This clarification is mentioned in the methods. This analysis treated the effect of flow on these variables as a trend; we discussed this issue with a biostatistician, who felt that there are no other statistical tests appropriate for this situation. We also provide the standard error on the mean for each curve, to address the reproducibility of the data. Additionally, we tested the difference between specific shear stress magnitudes in Figure 2—figure supplement 1, we performed a 2 way ANOVA test to compare the 3 conditions. The results validate the conclusion that fluid shear stress at different magnitudes induces statistically different responses.

*Also, the responses to points 12 and 13 about the MMP 9 expression are vague. The authors state that they could not observe any change in the MMP 9 intensity in HUVECs in the gradient chamber and conclude that “MMP9 analysis in vitro does not appear to be a fruitful line of investigation”, whereas there is a body of literature reporting in vitro studies of MMP 9 expression in various cell types. Similarly, they do not offer an explanation on how VEGFR3 expression regulates MMP9 expression*.

Regarding MMP9, available methods for analyzing its expression are limited to biochemistry (QPCR, zymography, etc.).These are not readily applicable to the gradient chamber where staining is the only relevant method for visualizing differences across the chamber.

Regarding VEGFR3, we do not think the relationship between VEGFR3 and MMP9 expression is direct. Changing the set point results in changes in many signaling pathways. Elucidating all of the consequences is clearly beyond the scope of this paper.

Moreover, we do not feel that elucidating the mechanisms that govern MMP9 expression in vitro is an essential part of this story as endothelial cells in vitro cannot remodel in a meaningful way.

Reviewer #3:

*The Coon et al. manuscript that has apparently been sent to another journal presents some complications*:

*1) There is overlap as in both manuscripts induction of VEGFR3 phosphorylation by FSS is now presented as a novel finding. The new*
Figure 5
*in the revised* eLife *manuscript is not of the necessary standard as it fails to show total VEGFR3 protein in the lysates, only P-VEGFR3 is displayed. The total VEGFR3 protein is analysed in a similar*
Figure 4
*of the other manuscript, but one cannot ask the* eLife *readers to consult another paper for such basic control*.

The other manuscript is now accepted and is cited to address this issue.

*2) The* eLife *manuscript avoids clearly mentioning the VEC;VEGFR2 complex, whereas the other manuscript claims VEGFR2;VEGFR3;VEC mechanosensory complex is involved, and speculates that a critical component may actually be a phosphatase*

*3) Whereas the* eLife *manuscript emphasizes VEGFR3 alone as the FSS set point, the other manuscript says in the abstract that “VEGFR2 and VEGFR3 signal redundantly downstream of VE-cadherin”*.

*4) Both manuscripts also show evidence of “significant” VEGFR3 expression in aortic endothelium. It is difficult to know if this amount is biologically significant*.

*5) The* eLife *abstract says: “VEGFR3 modulates arterial lumen diameter consistent with flow-dependent remodeling” (it should probably be “aortic”rather than “arterial” here, as other arteriae were not studied). The abstract of the other manuscript has a similar result: “VEGFR3 expression is observed in the aortic endothelium where it contributes to flow responses in vivo”*.

We do not understand what is the concern behind these comments that compare the two manuscripts. None of the data overlap and the main conclusions are completely distinct. The major issue flagged by the reviewers was that we invoked VEGFR3 as a component of the mechanosensory mechanism without providing the evidence. This work is now in press and cited in the current manuscript.

Regarding the biological significance of VEGFR3 in arterial endothelial cells, we demonstrated that its deletion results in dramatic remodeling of the aorta in two different organisms. It seems to us that this demonstrates biological significance. We also changed “arterial” by “aortic” in the new abstract, to reflect the reviewer’s concern.